# Gender Differences in the Epidemiological Characteristics and Long-Term Trends of Injuries in Taiwan from 1998 to 2015: A Cross-Sectional Study

**DOI:** 10.3390/ijerph19052531

**Published:** 2022-02-22

**Authors:** Pin-San Chou, Shi-Hao Huang, Ren-Jei Chung, Yao-Ching Huang, Chi-Hsiang Chung, Bing-Long Wang, Chien-An Sun, Shu-Min Huang, I-Long Lin, Wu-Chien Chien

**Affiliations:** 1Graduate Institute of Life Sciences, National Defense Medical Center, Taipei 11490, Taiwan; duke.zhou@gmail.com; 2Department of Chemical Engineering and Biotechnology, National Taipei University of Technology (Taipei Tech), Taipei 10608, Taiwan; rjchung@ntut.edu.tw; 3Department of Medical Research, Tri-Service General Hospital, National Defense Medical Center, Taipei 11490, Taiwan; g694810042@gmail.com; 4School of Public Health, National Defense Medical Center, Taipei 11490, Taiwan; billwang1203@gmail.com; 5Department of Public Health, College of Medicine, Fu-Jen Catholic University, New Taipei 242062, Taiwan; 040866@mail.fju.edu.tw; 6Big Data Research Center, College of Medicine, Fu-Jen Catholic University, New Taipei 242062, Taiwan; 7Department of Infection Control, Taipei Medical University Hospital, Taipei 11031, Taiwan; sharon6717@yahoo.com.tw; 8Department of Computer Science and Engineering, Tatung University, Taipei 104327, Taiwan; cyberpaul@gm.ttu.edu.tw; 9Taiwanese Injury Prevention and Safety Promotion Association (TIPSPA), Taipei 11490, Taiwan

**Keywords:** gender difference, injury, epidemiology, long-term trend analysis

## Abstract

Objective: This study used a long-term trend analysis to investigate whether gender differences were related to the risk of injury and epidemiological characteristics in Taiwan from 1998 to 2015. Materials and methods: Data on 4,647,259 hospitalized patients that were injured from 1 January 1998, to 31 December 2015 were collected from the National Health Insurance Research Database (NHIRD). Among the injured patients, 2,721,612 males and 1,925,446 females were identified. Patients were age-, gender-, and index date-matched. Multiple logistic regression was used to analyze the risks of injury via gender differences. A *p*-value < 0.05 was considered significant. Results: The injury risk of the male patients was 1.4 times higher than that of female patients (AOR = 1.427, 95% CI = 1.40–1.44). The rising trend of male injured hospitalized patients was also greater than that of female injured hospitalized patients. Conclusion: Males were more at risk of injury than females. Gender differences were related to the increased risk of epidemiological characteristics of injury.

## 1. Introduction

According to the Road Traffic Accident Injury Report issued by the World Health Organization (WHO) in 2020, approximately 1.35 million people die in road traffic accidents each year. Approximately 2000–5000 people suffer from non-fatal injuries, and many people become disabled [1]. Injury causes considerable economic losses to individuals, families, and the entire country [1], and is thus viewed as an important public health issue by governments around the world. Most related epidemiological studies have found an overall gender difference or age difference in injured patients [2]. In addition, injury is an important health problem in Taiwan. According to statistics from Taiwan’s Ministry of Health and Welfare, the mortality rate is observed to be higher in men than in women irrespective of injury type, but non-fatal injuries may show different results [3]. “Injury” also refers to unintentional events—such as transportation accidents, accidental poisoning, falls, fire, and accidental drowning—and deliberate injury events such as suicide and homicide [3].

According to statistics from the Ministry of Health and Welfare in Taiwan in 2018, the total number of deaths due to “accident injuries” reached 6846. “Accident injuries” rank sixth among the top 10 mortality causes in Taiwan. Among other causes, traffic accidents accounted for 3209 persons (46.9%), whereas falls accounted for 1409 persons (20.6%) [3].

Together, these top two causes of death accounted for 67.5% of deaths. Compared to 2008, the number of deaths caused by injury decreased by 231 persons (by 3.3%). Except for traffic accidents, which decreased by 17.1%, and accidental drowning, which decreased by 33.7%, all other categories showed an increasing trend [3].

In 2002, the WHO published “Gender and Road Traffic Accident Injury”, which stated that, globally, gender differences account for the largest injury death rate. Male deaths due to traffic accidents are almost three times (2.7) that of females [4]. In developing countries, male injury and death rates are relatively high in all types of road traffic accidents. Differences in gender (social/structural differences between men and women) and sex (biological, psychological, and environmental vulnerability differences) may be important factors that cause varying risk of injury [5]. In addition, injuries are the main cause of premature death. Men are more likely to die, and to die earlier, from diseases than women [6,7]. In all age groups in low to high-income countries, the injury, suicide, and homicide mortality rates of men are higher than those of women [8]. The only exception is in low-income and high-income countries, where the homicide rate of girls is higher or similar to that of boys in the killing of children under 15 years of age [9].

Although the issue of direct causality is still under debate, previous literature shows that gender differences are risk factors for occupational injuries. Do men suffer more occupational injuries than women, or vice versa [10]? Various studies by the United States (US) heavy industry and the Health Department of British Columbia in Canada concluded that women’s risk of occupational injuries is 1.58 times that of men [11,12]. Women also have a higher risk of upper-extremity musculoskeletal diseases than men, but this cannot be attributed solely to occupational injuries [13] or to increased workplace pressure [14]. In addition, power supply companies in Southern California and the American Institute of Architecture reported that women’s injury rate is greater than that of men [15,16]. In contrast, other studies in different countries such as Sweden and Taiwan concluded that the injury rate of men is greater than that of women [17,18].

To date, research on the relationship between gender and injury remains limited. Such research also has limitations, including small sample sizes, insufficient tracking time, physical and mental illnesses that are not controlled, and unexplored gender differences in injuries. Therefore, this study assumes that gender differences are related to injuries. Data on injured male and female patients from 1998 to 2015 were collected from the National Health Insurance Research Database (NHIRD) of Taiwan for a long-term follow-up analysis of the epidemiological characteristics of gender differences in relation to injury.

## 2. Materials and Methods

### 2.1. Data Source

Taiwan’s National Health Insurance System was implemented in 1995 and currently covers 99% of all citizens. The Data Science Center of the Ministry of Health and Welfare (HWDC, MOHW) collects all emergency room and hospitalization data. In addition, the law requires medical institutions to submit monthly declaration files for emergency room and hospitalization expenses. Therefore, the Data Science Center is the most authoritative data source for medical care-related research [19].

This study used a cross-sectional design to analyze inpatient medical declaration files from 1998 to 2015. The data from 1998 to 2015 were derived from the International Classification of Diseases, 9th Revision, Clinical Modification (ICD-9-CM), while those from 2016 to 2020 were based on the 10th clinical revision of the International Classification of Diseases (ICD-10-CM). The ICD-9-CM codes do not match those of the ICD-10-CM. All procedures carried out involving human participants complied with the ethical standards of the institution, the National Research Council, and the 1964 Declaration of Helsinki and its subsequent amendments, or similar ethical standards.

All methods were carried out following relevant guidelines and regulations. Informed consent was obtained from all participating subjects, including the parents and/or legal guardians of subjects under 18 years of age. This study used secondary data without any personally identifiable information and was approved by the National Defense Medical Center Tri-Service General Hospital Ethical Review Board (TSGHIRB 1-105-05-142), thereby waiving the requirement for individual written informed consent. Figure 1 shows the research flow chart.

### 2.2. Variable Definition

Variables include: gender (male, female), age (1–4, 5–14, 15–24, 25–44, 45–64, and 65 years and older), Charlson’s Comorbidity Index (CCI), intentionality of injury (International Classification of Diseases, 9th Revision, Clinical Modification (ICD-9-CM) E-Codes: E800.0–E949.5 unintentional, E950.0–E979.9 deliberate, and E980.0–E989.9 undetermined whether accidental or intentional injury), cause of injury (ICD-9-CM E-codes: E800.0–E848.9 transportation-related injuries, E850.0–E869.1 poisoning, E870.0–E879.9 medical accidents, E880.0–E888.9 falls, E890.0–E899.9 burns, E900.0–E909.9 natural and environmental factors, E910.0–E910.9 drowning, E911.0–E919.9 suffocation, E915.0–E916.9 and E920 crushing, cutting, and piercing, E921.0–E949.5 other unintentional injuries, E950.0–E959.9 suicide, E960.0–E979.9 homicide, and E980.0–E989.9 undetermined), low income (yes, no), major disease (yes, no), operation or not (yes, no), history of mental illness (yes, no), hospital level (medical center, regional hospital), degree of urbanization (high, medium, low), seasonality (spring, summer, autumn, winter), medical treatment types (internal medicine, surgery, gynecology, pediatrics, other departments), length of stay (days), medical expenses (NT$), and prognosis (survival rate, mortality). The CCI [10] states to select the patient diagnosis code (ICD-9-CM N-Code), weigh it according to the scoring criteria defined by Charlson, and calculate the total score. A high score indicates complications or a severe diagnosis. In addition, the prognosis of the injured includes death in the hospital and terminally ill patients’ voluntary discharge.

### 2.3. Statistical Analysis

The descriptive statistics of this study are presented in the form of percentages, averages, and standard deviations. A Chi-square test, Fisher exact test, Student’s t-test, and one-way ANOVA were used to assess the categorical and continuous variables between men and women. After adjusting for age, gender, and comorbidities, the CCI assesses comorbidity level by considering both the number and severity of 19 pre-defined comorbid conditions. A weighted score of a client’s comorbidities is then obtained and can be used to predict short- and long-term outcomes, such as function, length of hospital stay, and mortality rates. After controlling for the main reasons for admission and severity, survival analysis was used to explore the relationship between comorbidity and death within one year, and type 1 comorbidity was weighted according to the adjusted relative risk. The 10-year survival of the patient was verified. Table 1 shows the comorbidity categories and weights. If the relative risk is above 1.2 and less than 1.5, then the weight is 1; if the relative risk is above 1.5 and less than 2.5, the weight is 2; if the relative risk is above 2.5 and less than 3.5, then the weight is 3. The relative risk of type 2 comorbidity is greater than 6, which was used as its weight. We aggregated the comorbidity weights of patients, season, location, level of urbanization, and level of care. Adjusted odds ratio (AOR) controls for other predictor variables in a model, providing an idea of the dynamics between predictors. Multiple regression, which works with several independent variables, produces AORs. These and 95% confidence intervals (CIs) were calculated using a conditional logistic regression analysis to evaluate the effect of gender on the risk of accident injury. All analyses used SPSS Version 26 (IBM, Armonk, NY, USA). A *p*-value < 0.05 is considered statistically significant.

## 3. Results

This study collected data on 4,647,259 hospitalized injured patients in Taiwan from 1998 to 2015. Table 1 shows their basic characteristics. In terms of gender, 2,721,612 were male (58.57%) and 1,925,446 were female (41.43%). Among the patients, the average age was 43.56 ± 23.04 years old for males and 49.83 ± 24.14 years old for females. The highest hospitalization rate occurred between 25–44 years old (28.11%) for male patients and over 65 years old (31.69%) for females.

The proportion of male injured hospitalized patients was significantly higher than that of female injured hospitalized patients. Regarding the cause of injury, both men (37.35%) and women (38.23%) had the highest proportion of hospitalizations due to injuries from traffic accidents. In terms of the CCI, the scores of male patients were lower than those of females (0.46 ± 1.53 and 0.50 ± 1.64, respectively), which indicates a higher number and greater severity of injury complications for the latter.

In this study, a logistic regression was used to analyze the prognostic factors of injury and death. Table 2 shows that the risk of injury and death for male patients was 1.4 times that for female patients (AOR = 1.427, 95% CI = 1.40–1.44). The male patients over 65 years of age had a risk of injury and death that was 6.7 times (AOR = 6.703, 95% CI = 6.058–7.416) that for 5-year-old patients with injuries, while female patients over 65 years of age had a risk of injury and death that was 3.8 times (AOR = 3.803, 95% CI = 3.393–4.263) that for 5-year-old patients with injuries. In terms of intentionality of injury, the risk of injury and death was 2.1 times higher for male patients with unknown intent (AOR = 2.120, 95% CI = 6.058–7.416) and 2.8 times higher for female patients with unknown intent (AOR = 2.848, 95% CI = 2.684–3.023) than the risks for hospitalized patients with unintentional injuries. In terms of income, the risk of injury and death was 2.3 times higher for low-income male patients (AOR = 2.303, 95% CI = 2.196–2.416) and 2.4 times higher for female patients (AOR = 2.472, 95% CI = 2.293–2.666) than that for mid- or high-income patients. In terms of major diseases, the risk of injury and death was 3.4 times higher for male patients (AOR = 3.432, 95% CI = 3.361–3.505) and 3.5 times higher for female patients (AOR = 3.561, 95% CI = 3.464–3.661) than that for patients without major disease. In terms of psychiatric history, the risk of injury and death from mental illness among male patients was 1.6 times higher (AOR = 1.656, 95% CI = 1.625–1.688) than for those with non-psychiatric disease, whereas the risk of injury and death in female hospitalized patients suffering from mental illness was 1.5 times that for female patients suffering from sexual diseases (AOR = 1.555, 95% CI = 1.522–1.590). In terms of CCI, every increase in the score increases the risk of injury and death for male patients by 6%, and by 7.6% for female patients. This finding indicates that female injured hospitalized patients have concurrent injuries, and that symptoms are higher in number and more severe than those of men.

Table 3 and Figure 2 show the trend of injury hospitalization rates in Taiwan from 1998 to 2015. The total number of hospitalized patients was 212,241 (967.87 per 100,000 population) in 1998 and 323,588 (1377.43 per 100,000 population) in 2015. The hospitalization rate showed an upward trend and reached statistical significance. The number of male injured hospitalized patients was 127,911 (1137.65 per 100,000 population) in 1998 and 189,616 (1618.98 per 100,000 population) in 2015. The hospitalization rate showed an upward trend and reached statistical significance. The number of female injured hospitalized patients was 84,330 (789.22 per 100,000 population) in 1998 and 133,972 (1137.28 per 100,000 population) in 2015. The hospitalization rate showed an increasing trend and reached statistical significance.

## 4. Discussion

The results show that from 1998 to 2015 in Taiwan, the proportion of hospitalizations due to injury among men was significantly higher than that of women (58.57% vs. 41.43%). The CCI scores of male injured patients were lower than those of females (0.46 ± 1.53 and 0.50 ± 1.64), indicating that the number or severity of injury complications of the latter was higher or greater than that for men. The risk of death among male patients was 1.4 times that for females (AOR = 1.427, 95% CI = 1.40–1.44). In Taiwan, from 1998 to 2015, the increase in the risk of death of hospitalized male injured patients was greater than that for women (133/105). Injury is shown as the main cause of mortality in developing countries, and more than 90% of injury-related deaths worldwide occur when prevention efforts are insufficient [19,20]. Globally, the ratio of male to female injuries is 1.15–2:1 in developed countries and 2.6–9:1 in developing countries [21]. The sex ratio of harm in developed countries is much lower than that in developing countries [22]. At present, various studies on injury use population mortality as an important indicator of health. The injury mortality rate of men is approximately three times that of women. However, the important factors that cause the differences between male and female deaths require further research [23].

Men are born with a quantitative advantage, but this benefit diminishes over time. There are 105 boys for every 100 girls at birth [24]. However, the male fetal mortality rate is more than 7% higher than that of females [25]. The mortality gap widens immediately: by their first birthday, male fetal death rates are 21% higher than that of females [26]. The premature death of men continues until 65 years of age or older, at which point 75 men are counted for every 100 women over 65 years of age [27].

In 229 countries in the world, there are more male than female births; the sex ratio of males to females is 107:100 [27]. However, in most regions, men have not maintained their numerical advantage. Of the 229 countries, 93% more women than men over 65 years of age [27]. In addition, since 1970, the premature male mortality rate in 187 countries has led to a widening of the gender gap (the gap between adult males and females was 63 per 1000 in 1970 and increased to 80 per 1000 in 2010). This gender gap between the sexes has further increased [28].

Relevant literature shows that after infancy and before old age, men engage in more risk-taking behaviors, becoming prone to and experiencing more injuries, and dying of such causes more frequently [29]. For example, in the US, among young people aged 15–19, men are 2.5 times more likely to die from injuries than women. Similarly, the results of the present study show that the risk of death by injury in the hospital for men is 1.4 times that for women.

Why do men have a greater risk of death from injury than women? At present, the general belief is that because of socialized relationships, men have engaged in more dangerous behaviors than women since a young age and are less supervised by people who might protect them (to keep little boys from being harmed). Based on this current theory, attributing all risk-taking behaviors between men and women to gender differences is the only plausible explanation [29]. Observations on other primates and even other mammals show that sex differences are similar in injury and death patterns. The risk-taking behavior of primates is similar to that of humans [30,31,32,33]. Mammals have similar toy preferences to humans, indicating that differences in toy choices may reflect gender differences in activity preferences, rather than being mainly caused by socialization [31].

This is not a strange statement for chimpanzees, as female toddlers maintain a close relationship with the chimpanzee mother, while male toddlers wander, fight, and carry out dangerous activities. The reason is that the chimpanzee mother teaches the female toddler that, “Your brother can do this, but the beautiful little chimpanzee girl will not do that.” Even if it is a lion, we can also imagine that male lion cubs use their fathers as an example, and female cubs are taught otherwise. [29]. For example, female monkeys who received testosterone treatment during pregnancy exhibited “roughness” and “falling” afterward, intentional behaviors which were more similar to male monkeys. The treated female monkeys showed that the biological behaviors of primates and humans are very similar [34]. If primate theory applies to humans, then this does not mean that socialization is irrelevant in shaping human gender differences but is rather based on the biology that causes men and women to behave differently [29]. The aforementioned research shows that humans and primates have similar performances of gender differences in adventurous behaviors.

The biological differences between men and women affect the overall trend of behavior, the largest difference being levels of testosterone and estrogen. Each hormone has a different effect on the brain, e.g., in terms of driving, and each hormone provides its unique strengths and shortcomings [35].

A high testosterone level contributes to visual processing and spatial perception. Visual processing refers to the brain’s ability to understand the images recorded by the eyes, and better visual processing enables a driver to interpret road conditions and the movement of other vehicles more quickly [36]. When driving, one-tenth of a second, which may serve as the difference between an accident and avoidance thereof, makes visual processing an important ability [37]. Spatial awareness is the ability to understand the relationship between oneself and other objects. The effect of changes in spatial perception capabilities may be difficult to measure because many people do not know whether spatial misjudgment has caused their collision to a certain extent [38]. Testosterone also affects the temperament of drivers by increasing the level of aggression, which does not make men violent drivers, but rather, causes them to take greater risks on the road and to feel overconfident [39].

Estrogen has a unique effect on women’s driving style. First, estrogen can improve concentration, thereby increasing women’s ability to focus on the road without distractions [40]. Higher estrogen levels are also associated with improved memory, an overlooked attribute related to driving [40]. Higher estrogen levels can assist in recall of the required route or the time required for travel, often improving driver comfort and overall safety [40].

The difference between the benefits of estrogen and testosterone lies in their use: the concentration provided by estrogen and the improvement of memory often have no negative ramifications [41]. However, the risks of enhanced visual processing and effect of testosterone on behaviors can easily lead to dangerous outcomes [42,43].

In other possible situations, the accident injury is at times caused by the person themself. Alcoholism is an example. Alcohol consumption is common in accidental injuries and violence-related mortality [44]. Census surveys carried out in 10 countries indicate that the typical frequency and amount of drinking and the alcohol abuse rate are always greater for men than women [45]. Thus far, the largest predictor of accident injury among men is related to occupational alcoholism and negative behavior patterns [46]. For a long time, the belief is that masculinity may be harmful to men’s health; men take too many risks, which has a positive correlation with accident injury [47,48]. The integrated analysis of the previous 150 studies supports this hypothesis [49]. Current research suggests that 6-year-old boys are more adventurous than girls, but boys have a lower risk of injury than girls [50]. In addition, recent related research shows that male masculinity is more internalized in roles than for females, which explains the difference in male risk-taking by gender [51].

The differences in the structure and function of the brains of men and women are also of great scientific significance [52]. In general, male microglia are more common in specific brain areas (such as the hippocampus and cortex), and at ages three and 13 weeks. At 13 weeks, the soma is larger and seems to be more sensitive to stimulation or “ready to go” [52]. The “masculinity” of the brain depends on the activation of microglia [53,54]. The aforementioned research shows that the brain microglia structure and function and “masculinity” lead to differences in the performance of adventurous behavior between men and women. These studies indicate that risk-taking is seen as a relevant characteristic of men. In addition, risk-taking may be related to differences in gender competition [49]. In developed countries, women account for a large proportion of community workers and have to engage in high-risk injury behaviors similar to men (driving, outdoor work). In general, the smaller male–female injury mortality rate represents an increase in women’s participation in high-risk driving and outdoor work [55]. In Australia, the latest research points out that the gender difference between men and women in taking risks is gradually diminishing; women are becoming more adventurous, but this may not be conducive to the implementation of injury prevention work [56].

This study has several limitations. First, the data from the Data Science Center of the MOHW do not provide more important information about injuries. Second, information on other variables (such as daily drinking and smoking or biochemical test values) is also restricted. This study used E-Codes to analyze “accident injury”, but such information was only recorded in the hospitalization documents, and not in the outpatient and emergency documents. Therefore, the number of cases and overall medical utilization may be underestimated. Finally, due to the use of auxiliary database analysis in this study, the possibility of data classification errors and information bias cannot be ruled out.

## 5. Conclusions

This study confirms the correlation between gender difference and injury. The results show that the risk of injury and hospital death is greater for men than for women. Male patients have a greater risk of hospital death due to injuries than females.

A growing trend of fatal injuries is observed in both sexes, mainly due to the increase in traffic accidents. The possible solutions or interventions that can stop this negative trend are as follows:−Practice safe and alert driving: This practice includes getting enough sleep at night. While driving, avoid chatting, using mobile phones, or flipping through maps.−Maintain a safe driving distance: This includes the front and rear driving distance, lateral spacing, and overtaking distance.−Go out early, not in a hurry: On the way to or from work, avoid hurrying due to traffic jams and other factors.−Increase appropriate driving training: Training should include downhill driving, turning, avoiding improper braking, and other related aspects.−Be alert to changes in environmental conditions: These instances include late nights, cloudy and rainy days, dense fog, or road potholes.−Improve bad driving behavior: Avoid speeding, refusing to give each other courtesy, failing to let vehicles on the main lane go first on branch lanes, failing to maintain a safe distance, driving negligently, violating signal control, and failing to slow down.−Personal reasons for driving: These include drunk driving, taking psychedelic drugs and other similar controlled drugs, speeding, illegal parking, and failing to maintain a safe distance.

## Figures and Tables

**Figure 1 ijerph-19-02531-f001:**
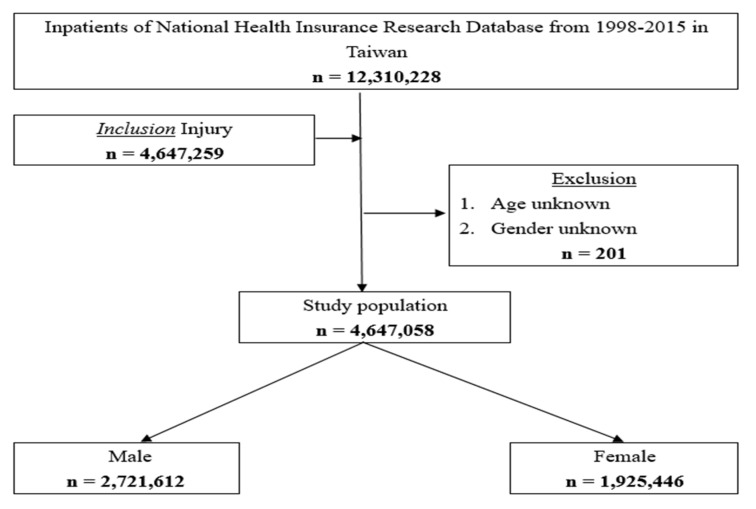
Research flow chart.

**Figure 2 ijerph-19-02531-f002:**
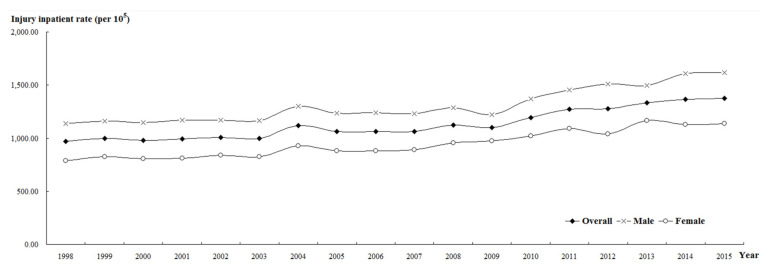
Trends in hospitalization rates for injuries in Taiwan from 1998 to 2015.

**Table 1 ijerph-19-02531-t001:** Demographic characteristics among injury patients.

Gender	Male		Female		*p*-Value
Variables	*n*	%	*n*	%	
	2,721,612	58.57	1,925,446	41.43	
**Age (mean ± SD, year)**	43.56 ± 23.04		49.83 ± 24.14		<0.001
**Age group**					<0.001
<5	104,815	3.85	77,595	4.03	
5–14	157,781	5.80	80,338	4.17	
15–24	442,543	16.26	225,069	11.69	
25–44	764,940	28.11	405,746	21.07	
45–64	660,428	24.27	526,470	27.34	
≧65	591,105	21.72	610,228	31.69	
**Type of injury**					<0.001
Traffic	692,564	37.35	503,348	38.23	<0.001
Poisoning	22,692	1.22	20,945	1.59	0.001
Medical-related	194,017	10.46	146,575	11.13	<0.001
Falls	386,429	20.84	377,222	28.65	<0.001
Burns and fires	6799	0.37	3192	0.24	0.290
Environment	19,927	1.07	11,628	0.88	0.102
Drowning	133,946	7.22	37,214	2.83	<0.001
Suffocation	12,147	0.66	7875	0.60	0.602
Adverse drug reaction	15,452	0.83	14,973	1.14	0.006
Other unintentional injuries	275,637	14.87	135,837	10.32	<0.001
Suicide	24,050	1.30	29,618	2.25	<0.001
Homicide/Abuse	59,161	3.19	16,964	1.29	<0.001
Intention unknown	11,196	0.60	11,259	0.86	0.022
**CCI**	0.46 ± 1.53		0.50 ± 1.64		<0.001

*p*: Chi-square/Fisher exact test on category variables and t-test on continuous variables, CCI: Charlson Comorbidity Index, AOR; Adjusted Odds Ratio; several patients did not provide information on the cause and intentionality of injury.

**Table 2 ijerph-19-02531-t002:** Injury factors of variables by using multivariable logistic regression.

Model	Male			Female		
Variables	AOR	95% CI	*p*-Value	AOR	95% CI	*p*-Value
**Gender**						
Male	1.427	1.40–1.44	<0.001			
Female	Reference					
**Age group**						
<5	Reference			Reference		
5–14	0.699	0.62–0.78	<0.001	0.758	0.65–0.87	<0.001
15–24	1.487	1.33–1.65	<0.001	1.093	0.96–1.23	0.152
25–44	2.146	1.93–2.37	<0.001	1.111	0.98–1.25	0.078
45–64	3.113	2.81–3.44	<0.001	1.627	1.45–1.82	<0.001
≧65	6.703	6.05–7.41	<0.001	3.803	3.39–4.26	<0.001
**Intentionality of injury**						
Unintentional	Reference			Reference		
Intentional	1.983	1.90–2.07	<0.001	2.848	2.68–3.02	<0.001
Intention unknown	2.120	1.94–2.31	<0.001	1.981	1.77–2.21	<0.001
**Low-income**						
Without	Reference			Reference		
With	2.303	2.19–2.41	<0.001	2.472	2.29–2.66	<0.001
**Catastrophic illness**						
Without	Reference			Reference		
With	3.432	3.36–3.50	<0.001	3.561	3.46–3.66	<0.001
**Psychiatric history**						
Without	Reference			Reference		
With	1.656	1.62–1.68	<0.001	1.555	1.52–1.59	<0.001
CCI_R	1.060	1.05–1.06	<0.001	1.076	1.07–1.08	<0.001

AOR = Adjusted Odds Ratio: Adjusted variables listed in the table, CI = Confidence Interval, Nagelkerke R-square = 0.172 (overall), 0.167 (male), 0.178 (female) CCI = Charlson Comorbidity Index; several patients did not provide information on the cause and intentionality of injury.

**Table 3 ijerph-19-02531-t003:** Injury hospitalization rate trends.

Gender	Male			Female		
Year	Inpatient	Mid-Year Population	Rate (per 10^5^)	Inpatient	Mid-Year Population	Rate (per 10^5^)
1998	127,911	11,243,408	1137.65	84,330	10,685,183	789.22
1999	131,412	11,312,728	1161.63	89,098	10,779,659	826.54
2000	130,567	11,392,050	1146.12	87,570	10,884,622	804.53
2001	133,826	11,441,651	1169.64	89,167	10,963,917	813.28
2002	134,662	11,485,409	1172.46	92,630	11,035,367	839.39
2003	134,256	11,515,062	1165.92	91,359	11,089,488	823.83
2004	150,243	11,541,585	1301.75	103,215	11,147,537	925.90
2005	142,933	11,562,440	1236.18	98,966	11,207,943	883.00
2006	143,966	11,591,707	1241.97	99,485	11,284,820	881.58
2007	142,802	11,608,767	1230.12	101,156	11,349,593	891.27
2008	149,499	11,626,351	1285.86	108,947	11,410,680	954.78
2009	141,952	11,636,734	1219.86	112,005	11,483,038	975.40
2010	159,399	11,635,225	1369.97	117,500	11,526,898	1019.35
2011	169,250	11,645,674	1453.33	126,270	11,579,238	1090.49
2012	176,373	11,673,319	1510.91	121,110	11,642,503	1040.24
2013	175,044	11,684,674	1498.06	136,082	11,688,843	1164.20
2014	187,899	11,697,971	1606.25	132,584	11,735,782	1129.74
2015	189,616	11,712,047	1618.98	133,972	11,780,027	1137.28
**Trend test**						
b (slope)	3,515.872	23,803.955	27.724	3,136.784	63,137.988	22.434
*p*-Value	<0.001	<0.001	<0.001	<0.001	<0.001	<0.001
R-square	0.865	0.890	0.835	0.937	0.992	0.909

## Data Availability

The NHIRD data that are used to support the findings of this study are restricted by TSGHIRB number 1-105-05-142 and the “Personal Information Protection Act” to protect patient privacy.

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
