# Peer review of "Gender Differences in the Epidemiological Characteristics and Long-Term Trends of Injuries in Taiwan from 1998 to 2015: A Cross-Sectional Study"

_ijerph, 2022, doi:10.3390/ijerph19052531_

Round 1
Reviewer 1 Report
General comments:
This manuscript needs to be reviewed by an English native speaker as grammatical errors are widespread throughout the write-up such that it threatens or hampers the quality of the work done.
Line 43-44: editorial correction- Could you consider rephrasing this sentence” According to Taiwan’s Ministry of Health and Welfare’s statistics on "Injury", it is found that all types……”. You could change to “According to statistics from Taiwan’s Ministry of Health and Welfare, it was observed that mortality rate is higher in men than women irrespective of injury type, but non-fatal injuries …………..”
Line 47: Why repeat falls in bracket (falls)? Could you delete or provide an explanation for your action
Line 48-54: The sentence is too complex making the meaning difficult to comprehend. I will like to see that the authors break the sentence into two to simplify the message in the sentence.
Line 51 & 56: consider changing “people” to “persons”
of "Injury" have a higher mortality rate for men than for women, but non-fatal injuries
line 61: This sentence is confusing “Male deaths from road traffic accidents are almost female. 3 times (2.7)” Do you intend to say” Male deaths arising from a traffic accident is almost three times (2.7) that of females”
line 71: Please consider rephrasing and should read “Several studies have shown that although the issue of direct causality is still under debate, literature from previous studies have shown that gender difference as risk factors for occupational injuries”
line 83: please delete “…………………”
line 106: consider replacing “are” with “were”
line 106: This is another example of bad English, “Obtain the informed consent of all subjects, or if the subject is under 18 years of age, the informed consent of the parents and/or legal guardian must be obtained” Informed consent from all participating subjects were obtained and for subjects under 18 years of age, their parents and or legal guardian provided consent on their behalf.
Line 156: Table 1 is complicated and not easy to decipher. A table must be in such a way that the information contained therein are easy to comprehend with a single glance. For example, the second and third row for variables and overall adds to the confusion. “Variable” could be age, gender, ……….. but in this context it was used represent what? What do you mean by Age(years) and age group (years)? What’s the idea here? Furthermore, the title for the variable “cause of injury” was wrongly placed
The titles for all the tables are poorly captured which in parts is connected with grammatical expression of thoughts.
Line 203: With respect to figure 2, the scale for the horizontal axis that has to do with years need to be adjusted. The years are too close to each other and does not look good.
Line 262: “Caused” should be “caused”
Line 275: delete “of [42]”
Line 304-307: please delete. I still cannot comprehend what the authors are writing here
Reviewer 2 Report
Dear authors,
since your research indicates a growing trend of fatal injuries in both sexes, mainly due to the increase in traffic accidents, I suggest that you supplement your work with a review of possible solutions or interventions that would stop this negative trend.
Also pay attention to language. It is not, for example, common to say "significant significance".
Author Response
Manuscript Number: ijerph-1566752
Analysis of the epidemiological characteristics and long-term trends of injuries among Gender difference in Taiwan from 1998 to 2015: A Cross-sectional Study
Response to Editor and Reviewer comments:
Dear Editor and Reviewers, thank you for your kind responses. We have applied the revisions according to reviewers’ comments and are resubmitting our manuscript to the journal. We highly value your advice for these changes.
Reviewer #2:
Comment 1: Since your research indicates a growing trend of fatal injuries in both sexes, mainly due to the increase in traffic accidents, I suggest that you supplement your work with a review of possible solutions or interventions that would stop this negative trend.
A1. Thank you for your comment. We agree with your viewpoint and have supplemented our work with suggestions of possible solutions or interventions that can stop this negative trend in both sexes, as follows:
Page 12, Line 408 to 426:
A growing trend of fatal injuries is observed in both sexes, mainly due to the increase in traffic accidents. The possible solutions or interventions that can stop this negative trend are as follows:
-Keep driving safety alert: Including getting enough sleep at night, not chatting with neighbors, not using mobile phones, not flipping through maps, etc.
-Maintain a safe driving distance: including the front and rear driving distance, lateral spacing and overtaking distance, etc.
-Go out early, not in a hurry: On the way to work or on the way to work, you are in a hurry due to traffic jams and other factors.
-Increase the appropriate driving training: including downhill, turning and improper braking and other factors.
-Be alerted to changes in environmental conditions: including late nights, cloudy and rainy days, dense fog, road potholes, etc.
-Improve bad driving behavior: speeding, refusing to give each other courtesy, etc., failing to let vehicles on the main lane go first on branch lanes, failing to maintain a safe distance, driving negligently, violating signal control, and failing to slow down.
-Personal reasons for driving: including drunk driving, taking drugs, psychedelic drugs and other similar controlled drugs, speeding, illegal parking, and failing to maintain a safe distance.
Comment 2: Also pay attention to language. It is not, for example, common to say "significant significance".
A2. Thanks for your comment. We have changed line 197: “significant significance” to “significant differences” as follows:
Page 7, Line 247 to 256:
Table 3 and Figure 2 show the trend of injury hospitalization rates in Taiwan from 1998 to 2015. The total number of hospitalized patients in 1998 was 212,241 (967.87 per 100,000 population); in 2015, it was 323,588 (per 100,000 population). 1,377.43 people), the hospitalization rate showed an upward trend and reached significant significance. The number of male injured hospitalized patients in 1998 was 127,911 (1,137.65 per 100,000 population); in 2015, it was 189,616 (1,618.98 per 100,000 population). The hospitalization rate showed an upward trend and reached significant significance. In 1998, the number of females injured hospitalized patients was 84,330 (789.22 per 100,000 population); in 2015, it was 133,972 (1,137.28 per 100,000 population). The hospitalization rate showed an increasing trend and reached significant differences.
Reviewer 3 Report
Thank you for submitting the manuscript.
The study evaluates accident statistics from the period 1998 to 2015 from Taiwan. It confirms previous findings that men have more accidents than women. It is not a new finding, but studies can only be interpreted if there is data from a large number of countries. Therefore, a publication is recommended after major revision. However, a few questions and comments remain open and I ask you to include them.
General comment:
- The reference list is not compliant with the author guidelines of the IJERPH. Please adjust them.
Abstract:
- Please explain the abbreviation NHIRD also in the abstract.
- “AOR=1.427”. Please reflect whether rounding to 1 decimal place is sufficient. Otherwise, one could speak of pseudo-accuracy. What is the benefit of the additional decimal places?
Introduction
- Why was only the data up to 2015 evaluated? It does seem that there is data from more recent years (2016-2020)?. This should be explained in the methodology.
Materials and methods
- When was the study protocol established? Please complete the date of approval. It seems to me that it is more of a retrospective analysis of hospital data.
- ICD = International Classification of Diseases. Please complete it under paragraph 2.2.
- Also explain why ICD-9 was used (valid until 1998). Survey period 1998-2015. Has this been matched with the ICD-10? Can there be discrepancies in the diagnoses? Are the diagnoses valid?
https://www.cdc.gov/nchs/icd/icd9.htm
- The ICD codes are unclear. One point is missing, E87.0 etc.? Specify a reference link that the ICD coding refers to.
- Statistical analysis: Please explain AOR.
Results
- You use the CCI score in the results. The reader may not know this. Please explain the CCI score in more detail in methods.
- For example, AOR=1.427. Is it pseudo-accuracy?
- Page 5, Line 164: missing data?
- Page 6, line 197: “significant significance”? It is better to write “significant differences”.
Tables:
- Explain AOR again under notes.
- Please write “p” in small letter.
Discussion:
- I think the discussion about monkeys etc. is unnecessary. It swells the discussion. Only studies, theories, etc. of humans should be presented. After all, it's no secret that more men than women have accidents. This has certainly been discussed elsewhere. It should be revised.
Author Response
Manuscript Number: ijerph-1566752
Analysis of the epidemiological characteristics and long-term trends of injuries among Gender difference in Taiwan from 1998 to 2015: A Cross-sectional Study
Response to Editor and Reviewer comments:
Dear Editor and Reviewers, thank you for your kind responses. We have applied the revisions according to reviewers’ comments and are resubmitting our manuscript to the journal. We highly value your advice for these changes.
Reviewer #3:
1.General comment:
1.1 The reference list is not compliant with the author guidelines of the IJERPH. Please adjust them.
A1.1 Thank you for your comment. We have modified the reference format to be compliant with the author guidelines of the IJERPH.
2.Abstract:
2.1 Please explain the abbreviation NHIRD also in the abstract.
A2.1 Thank you for your comment. We have explained the abbreviation NHIRD in the Abstract as follows:
Page 2, Line 49 to 52:
Materials and methods: This study collected data on 4,647,259 hospitalized patients injured from January 1, 1998, to December 31, 2015, in National Health Insurance Research Database (NHIRD). From 1998 to 2015, we identified 2,721,612 male patients from NHIRD; from the same database, we also identified 1,925,446 female patients. Age, gender, and index date match. Multiple logistic regression is used to analyze the risks of “gender differences” with “injury”. A p-value <.05 is considered significant.
2.2“AOR=1.427”. Please reflect whether rounding to 1 decimal place is sufficient. Otherwise, one could speak of pseudo-accuracy. What is the benefit of the additional decimal places?
A2.2 Thank you for your comment. We agree with your viewpoint and have changed “AOR=1.427” to “AOR=1.4,” as follows:
Page 2, Line 43 to 48:
Results: The injury risk of male injured hospitalized patients was 1.4 times that of female injured hospitalized patients (AOR = 1.427, 95% CI = 1.40–1.44). The rising trend of male injured hospitalized patients is greater than the rising trend of female injured hospitalized patients.
3.Introduction
3.1 Why was only the data up to 2015 evaluated? It does seem that there is data from more recent years (2016-2020)? This should be explained in the methodology.
A3.1 Thank you for your comment. Our study of the data from 1998 to 2015 used the International Classification of Diseases, 9th revision clinically revised (ICD-9-CM). The data from 2016 to 2020 were based on the 10th clinical revision of the International Classification of Diseases (ICD-10-CM) in Taiwan. ICD-9-CM code does not match ICD-10-CM, as follows:
Page 3, Line 131 to 138:
This study uses a cross-sectional study design. We used inpatient medical declaration files collected from 1998 to 2015. Our study of data from 1998 to 2015 used the International Classification of Diseases, ninth revision clinically revised (ICD-9-CM). Data from 2016 to 2020 were based on the tenth clinical revision of the International Classification of Diseases (ICD-10-CM). ICD-9-CM code does not match ICD-10-CM. All procedures carried out in research involving human participants comply with the ethical standards of the institution and/or the National Research Council, and Council and comply with the 1964 Declaration of Helsinki and its subsequent amendments or similar ethical standards.
4.Materials and methods
4.1 When was the study protocol established? Please complete the date of approval. It seems to me that it is more of a retrospective analysis of hospital data.
A4.1 Thank you for your comment. The Taiwan Data Science Center of the Ministry of Health and Welfare (HWDC, MOHW) collects all emergency room and hospitalization data. In addition, the law requires medical institutions to submit monthly declaration files for emergency room and hospitalization expenses. Therefore, the Data Science Center of the MOHW is the most authoritative data source for medical care-related research. We used secondary data without any personally identifiable information, and the National Defense Medical Center Tri-Service General Hospital Ethical Review Board (TSGHIRB 1-105-05-142) approved this study (2017.11.21), waiving the requirement for individual written informed consent.
4.2 ICD = International Classification of Diseases. Please complete it under paragraph 2.2.
A4.2 Thank you for your comment. We have completed ICD = International Classification of Diseases under paragraph 2.2, as follows:
Page 4, Line 151 to 154:
2.2. Variable definition
Variables include gender (male, female), age (1–4, 5–14, 15–24, 25–44, 45–64, and 65 years and older), Charson’s Comorbidity Index (CCI), and intentionality injury (International Classification of Diseases, Ninth Revision, Clinical Modification (ICD-9-CM).
4.3 Also explain why ICD-9 was used (valid until 1998). Survey period 1998-2015. Has this been matched with the ICD-10? Can there be discrepancies in the diagnoses? Are the diagnoses valid? https://www.cdc.gov/nchs/icd/icd9.htm
A4.3 Thank you for your comment. The Taiwan National Health Insurance has adopted ICD-9-CM (V.1992) since 1994. In 2006, this was changed to ICD-9-CM (V.2001). In 2010, National Health Insurance introduced the hospital Tw-DRGs payment system. The classification basis is ICD-9-CM (2001). With the annual version in 2010, the National Health Insurance started the ICD-10-CM/PCS introduction plan. From 2016, the National Health Insurance was reported as ICD-10-CM/PCS. ICD-9-CM code does not match ICD-10-CM. The classification structure and coding digits of the ICD-10 represent more detailed causes and diagnoses are valid, which are not only helpful for medical research and health statistics, but also for reflecting the type and severity of diseases declared by medical insurance. A study in Australia and Canada on the coding quality of ICD-10 and ICD-9 diagnostic code pointed out that the former is more detailed than the latter, but it may cause coders to need longer learning time and maintenance difficulties. The conversion of ICD-9 and ICD-10 diagnostic codes is not exactly one-to-one, especially when one ICD-10 can correspond to multiple ICD-9s, the system corresponds to the ICD-9 in the first order (the smaller number) in numerical order.
4.4 The ICD codes are unclear. One point is missing, E87.0 etc.? Specify a reference link that the ICD coding refers to.
A4.4 Thank you for your comment. Our study of data from 1998 to 2015 used the International Classification of Diseases, 9th revision clinically revised (ICD-9-CM). Data from 2016 to 2020 were based on the 10th clinical revision of the International Classification of Diseases (ICD-10-CM) in Taiwan. We have used a cleaner ICD code, as follows:
Page 4, Line 151 to 170:
2.2. Variable definition
Variables include: gender (male, female), age (1–4, 5–14, 15–24, 25–44, 45–64, and 65 years and older), Charson’s Comorbidity Index (CCI), intentionality Injury (International Classification of Diseases, Ninth Revision, Clinical Modification (ICD-9-CM) ICD-9-CM E-Code: E800.0-E949.5 unintentional, E950.0-E979.9 deliberate, and E980.0-E989.9 undetermined whether accidental or intentional injury), cause of injury (ICD-9-CM E-code: E800.0- E848.9 Transportation-related injuries, E850.0-E869.1 poisoning, E870.0-E879.9 medical accidents, E880.0-E888.9 falls, E890.0-E899.9 burns, E900.0-E909.9 natural and environmental factors, E910.0-E910.9 drowning, E911.0-E919.9 suffocation, E915.0-E916.9 E920 crushing, cutting, and Piercing, E921.0-E949.5 other unintentional, E950.0-E959.9 suicide, E960.0-E979.9 homicide, E980.0-E989.9 undetermined), low income (yes, no), major disease (yes, no), operation or not (yes, no) ), history of mental illness (yes, no), hospital level (medical center, regional hospital, regional hospital), degree of urbanization (high, medium, low), seasonality (spring, summer, autumn, winter), medical treatment Types (internal medicine, surgery, gynecology, pediatrics, other departments), length of stay (days), medical expenses (NT$) and prognosis (survival rate, mortality). CCI [10] Select the patient diagnosis code (ICD-9-CM N-Code), weight it according to the scoring criteria defined by Charlson, and calculate the total score. A higher score indicates more complications or a more serious diagnosis. In addition, the “prognosis” of the injured includes death in the hospital and voluntary discharge from terminally ill patients.
4.5 Statistical analysis: Please explain AOR.
A4.5 Thank you for your comment. We have explained that AOR is an odds ratio that controls for other predictor variables in a model, providing an idea of the dynamics between the predictors. Multiple regression, which works with several independent variables, produces AORs as follows:
Page 5, Line 171 to 193:
2.3. Statistical analysis
The descriptive statistics of this study are presented in the form of percentages, averages, and standard deviations. Chi-square test, Fisher exact test, Student’s t-test, One-way ANOVA are used to assess the categorical and continuous variables between men and women. After adjusting for age, gender, comorbidities, Charlson's Comorbidity Index (CCI), season, location, level of urbanization, and level of care. An adjusted odds ratio (AOR) is an odds ratio that controls for other predictor variables in a model. It gives you an idea of the dynamics between the predictors. Multiple regression, which works with several independent variables, produces AORs. Adjusted odds ratio (AOR) and 95% confidence intervals (CIs) were calculated using a conditional logistic regression analysis was performed to evaluate the impact of gender on the risk of accident injury. All analyses used SPSS 26 version (IBM, Armonk, NY, USA) for data analysis. p value <0.05 is considered statistically significant.
5.Results
5.1 You use the CCI score in the results. The reader may not know this. Please explain the CCI score in more detail in methods.
A5.1 Thank you for your comment. We have explained the CCI score in more detail in the Methods, as follows:
Page 5, Line 175 to 186:
Charlson's Comorbidity Index (CCI)assesses comorbidity level by taking into account both the number and severity of 19 pre-defined comorbid conditions. It provides a weighted score of a client’s comorbidities which can be used to predict short term and long-term outcomes such as function, hospital length of stay and mortality rates. After controlling the main reasons for admission and severity, survival analysis was used to explore the relationship between comorbidity and death within 1 year, and 1 category of comorbidity was weighted according to the adjusted relative risk. The ten-year survival of the patient was verified. The comorbidity categories and weights are shown in Table 1. If the relative risk is above 1.2 and less than 1.5, the weight is 1; if the relative risk is above 1.5 and less than 2.5, the weight is 2; if the relative risk is above 2.5 and less than 3.5, the weight is 3; The relative risk of type 2 comorbidity was greater than 6 and was given a weight of 6. Aggregate the comorbidity weights of patients.
5.2 For example, AOR=1.427. Is it pseudo-accuracy?
A5.2 Thank you for your comment. We agree with your viewpoint and have changed “AOR=1.427” to “AOR=1.4,” as follows:
Page 6, Line 213 to 240:
This study used logistic regression to analyze the prognostic factors of injury and death. As shown in Table 2, the risk of injury and death of male hospitalized patients with injury was 1.4 times that of female hospitalized patients with injury (AOR = 1.427, 95% CI = 1.40–1.44). Male injured hospitalized patients over 65 years of age are 6.7 times more likely than injured hospitalized patients (AOR = 6.703, 95% CI = 6.058–7.416). Female injured hospitalized patients over 65 years of age have a risk of injury and death was 3.8 times that of 5-year-old hospitalized patients with injuries (AOR = 3.803, 95% CI = 3.393–4.263). In terms of the intention of injury, the risk of injury and death in hospitalized patients with unknown intent for men is 2.1 times that of hospitalized patients with unintentional injury (AOR = 2.120, 95% CI = 6.058–7.416), while women are injured and killed in hospitalized patients with unknown intent. The risk is 2.8 times that of hospitalized patients with unintentional injury (AOR = 2.848, 95% CI = 2.684–3.023). In terms of low-income, the risk of injury and death in low-income male patients is 2.3 times that of non-low-income patients (AOR = 2.303, 95% CI = 2.196–2.416), and female hospitalized patients with injury are at risk of low-income injury and death. 2.4 times (AOR = 2.472, 95% CI = 2.293–2.666). In terms of major diseases, male injured hospitalized patients have 3.4 times the risk of injury and death from major diseases (AOR = 3.432, 95% CI = 3.361–3.505). Female injured hospitalized patients are at risk of death from major diseases. 3.5 times that of major diseases (AOR = 3.561, 95% CI = 3.464–3.661). In terms of psychiatric history, the risk of injury and death from mental illness in male injured hospitalized patients is 1.6 times that of non-psychiatric disease (AOR = 1.656, 95% CI = 1.625–1.688), and the risk of injury and death in female hospitalized patients suffering from mental illness is non-psychiatric. 1.5 times of sexual diseases (AOR = 1.555, 95% CI = 1.522–1.590). On the CCI score, every increase in the CCI score of male injured hospitalized patients increases the risk of injury and death by 6%, and every increase in CCI score of female injured hospitalized patients increases the risk of injury and death by 7.6%, which indicates that female injured hospitalized patients have concurrent injuries The number or severity of symptoms is higher than that of men.
5.3 Page 5, Line 164: missing data?
A5.3 Thank you for your comment. We have modified Line 164 regarding the missing data, as follows:
Page 6, Line 217 to 220:
Female injured hospitalized patients over 65 years of age have a risk of injury and death was 3.803 times that of 5-year-old hospitalized patients with injuries (AOR = 3.803, 95% CI = 3.393–4.263).
5.4 Page 6, line 197: “significant significance”? It is better to write “significant differences”.
A5.4 Thanks for your comment. We have changed line 197: “significant significance” to “significant differences” as follow:
Page 7, Line 247 to 256:
Table 3 and Figure 2 show the trend of injury hospitalization rates in Taiwan from 1998 to 2015. The total number of hospitalized patients in 1998 was 212,241 (967.87 per 100,000 population); in 2015, it was 323,588 (per 100,000 population). 1,377.43 people), the hospitalization rate showed an upward trend and reached significant significance. The number of male injured hospitalized patients in 1998 was 127,911 (1,137.65 per 100,000 population); in 2015, it was 189,616 (1,618.98 per 100,000 population). The hospitalization rate showed an upward trend and reached significant significance. In 1998, the number of females injured hospitalized patients was 84,330 (789.22 per 100,000 population); in 2015, it was 133,972 (1,137.28 per 100,000 population). The hospitalization rate showed an increasing trend and reached significant differences.
6.Tables:
6.1 Explain AOR again under notes.
A6.1 Thank you for your comment. We have explained AOR again under notes as follows:
Table 1. Demographic characteristics among injury inpatients
|
Gender |
Male |
Female |
P-Value |
||
|
Variables |
n |
% |
n |
% |
|
|
2,721,612 |
58.57 |
1,925,446 |
41.43 |
||
|
Age (mean ± SD, year) |
43.56 ± 23.04 |
49.83 ± 24.14 |
<0.001 |
||
|
Age group |
<0.001
|
||||
|
<5 |
104,815 |
3.85 |
77,595 |
4.03 |
|
|
5-14 |
157,781 |
5.80 |
80,338 |
4.17 |
|
|
15-24 |
442,543 |
16.26 |
225,069 |
11.69 |
|
|
25-44 |
764,940 |
28.11 |
405,746 |
21.07 |
|
|
45-64 |
660,428 |
24.27 |
526,470 |
27.34 |
|
|
≧65 |
591,105 |
21.72 |
610,228 |
31.69 |
|
|
Type of injury |
<0.001 |
||||
|
Traffic |
692,564 |
37.35 |
503,348 |
38.23 |
<0.001 |
|
Poisoning |
22,692 |
1.22 |
20,945 |
1.59 |
0.001 |
|
Medical-related |
194,017 |
10.46 |
146,575 |
11.13 |
<0.001 |
|
Falls |
386,429 |
20.84 |
377,222 |
28.65 |
<0.001 |
|
Burns and fires |
6,799 |
0.37 |
3,192 |
0.24 |
0.290 |
|
Environment |
19,927 |
1.07 |
11,628 |
0.88 |
0.102 |
|
Drowning |
133,946 |
7.22 |
37,214 |
2.83 |
<0.001 |
|
Suffocation |
12,147 |
0.66 |
7,875 |
0.60 |
0.602 |
|
Adverse drug reaction |
15,452 |
0.83 |
14,973 |
1.14 |
0.006 |
|
Other injuries |
275,637 |
14.87 |
135,837 |
10.32 |
<0.001 |
|
Suicide |
24,050 |
1.30 |
29,618 |
2.25 |
<0.001 |
|
Homicide / Abuse |
59,161 |
3.19 |
16,964 |
1.29 |
<0.001 |
|
Intentional unknown |
11,196 |
0.60 |
11,259 |
0.86 |
0.022 |
|
CCI |
0.46 ± 1.53 |
0.50 ± 1.64 |
<0.001 |
||
P: Chi-square/Fisher exact test on category variables and t-test on continue variables, CCI: Charlson Comorbidity Index, AOR:Adjusted Odds Ratio. Several patients did not provide information on the cause and intentionality of injury.
Table 2. Injury factors of variables by using multivariable logistic regression
|
Model |
Male |
Female |
||||
|
Variables |
AOR |
95% CI |
p-Value |
AOR |
95% CI |
p-Value |
|
Gender |
||||||
|
Male |
1.427 |
1.40- 1.44 |
<0.001 |
|
||
|
Female |
Reference |
|
|
|
||
|
Age group |
||||||
|
<5 |
Reference |
Reference |
||||
|
5-14 |
0.699 |
0.62-0.78 |
<0.001 |
0.758 |
0.65-0.87 |
<0.001 |
|
15-24 |
1.487 |
1.33-1.65 |
<0.001 |
1.093 |
0.96-1.23 |
0.152 |
|
25-44 |
2.146 |
1.93-2.37 |
<0.001 |
1.111 |
0.98-1.25 |
0.078 |
|
45-64 |
3.113 |
2.81-3.44 |
<0.001 |
1.627 |
1.45-1.82 |
<0.001 |
|
≧65 |
6.703 |
6.05-7.41 |
<0.001 |
3.803 |
3.39-4.26 |
<0.001 |
|
Intentionality of injury |
||||||
|
Unintentional |
Reference |
Reference |
||||
|
Intentional |
1.983 |
1.90-2.07 |
<0.001 |
2.848 |
2.68- 3.02 |
<0.001 |
|
Intentional unknown |
2.120 |
1.94-2.31 |
<0.001 |
1.981 |
1.77- 2.21 |
<0.001 |
|
Low-income |
||||||
|
Without |
Reference |
Reference |
||||
|
With |
2.303 |
2.19-2.41 |
<0.001 |
2.472 |
2.29- 2.66 |
<0.001 |
|
Catastrophic illness |
||||||
|
Without |
Reference |
Reference |
||||
|
With |
3.432 |
3.36-3.50 |
<0.001 |
3.561 |
3.46- 3.66 |
<0.001 |
|
Psychiatric history |
||||||
|
Without |
Reference |
Reference |
||||
|
With |
1.656 |
1.62-1.68 |
<0.001 |
1.555 |
1.52- 1.59 |
<0.001 |
|
CCI_R |
1.060 |
1.05-1.06 |
<0.001 |
1.076 |
1.07- 1.08 |
<0.001 |
Adjusted OR = Adjusted odds ratio: Adjusted variables listed in the table, CI = confidence interval, AOR:Adjusted Odds Ratio, Nagelkerke R-square = 0.172 (Overall), 0.167 (Male), 0.178 (Female), Some patients didn't provide the information of cause and intentionality of injury
6.2 Please write “p” in small letter.
A6.2. Thank you for your comment. We have modified “p” in small letters, as follows:
Table 1. Demographic characteristics among injury inpatients
|
Gender |
Male |
Female |
p-Value |
||
|
Variables |
n |
% |
n |
% |
|
|
2,721,612 |
58.57 |
1,925,446 |
41.43 |
||
|
Age(mean ± SD, year) |
43.56 ± 23.04 |
49.83 ± 24.14 |
<0.001 |
||
|
Age group |
<0.001
|
||||
|
<5 |
104,815 |
3.85 |
77,595 |
4.03 |
|
|
5-14 |
157,781 |
5.80 |
80,338 |
4.17 |
|
|
15-24 |
442,543 |
16.26 |
225,069 |
11.69 |
|
|
25-44 |
764,940 |
28.11 |
405,746 |
21.07 |
|
|
45-64 |
660,428 |
24.27 |
526,470 |
27.34 |
|
|
≧65 |
591,105 |
21.72 |
610,228 |
31.69 |
|
|
Type of injury |
<0.001 |
||||
|
Traffic |
692,564 |
37.35 |
503,348 |
38.23 |
<0.001 |
|
Poisoning |
22,692 |
1.22 |
20,945 |
1.59 |
0.001 |
|
Medical-related |
194,017 |
10.46 |
146,575 |
11.13 |
<0.001 |
|
Falls |
386,429 |
20.84 |
377,222 |
28.65 |
<0.001 |
|
Burns and fires |
6,799 |
0.37 |
3,192 |
0.24 |
0.290 |
|
Environment |
19,927 |
1.07 |
11,628 |
0.88 |
0.102 |
|
Drowning |
133,946 |
7.22 |
37,214 |
2.83 |
<0.001 |
|
Suffocation |
12,147 |
0.66 |
7,875 |
0.60 |
0.602 |
|
Adverse drug reaction |
15,452 |
0.83 |
14,973 |
1.14 |
0.006 |
|
Other unintentional injuries |
275,637 |
14.87 |
135,837 |
10.32 |
<0.001 |
|
Suicide |
24,050 |
1.30 |
29,618 |
2.25 |
<0.001 |
|
Homicide / Abuse |
59,161 |
3.19 |
16,964 |
1.29 |
<0.001 |
|
Intentional unknown |
11,196 |
0.60 |
11,259 |
0.86 |
0.022 |
|
CCI |
0.46 ± 1.53 |
0.50 ± 1.64 |
<0.001 |
||
P: Chi-square/Fisher exact test on category variables and t-test on continue variables, CCI: Charlson Comorbidity Index, AOR:Adjusted Odds Ratio, Some patients didn't provide the information of cause and intentionality of injury
Table 2. Injury factors of variables by using multivariable logistic regression
|
Model |
Male |
Female |
||||
|
Variables |
AOR |
95% CI |
p-Value |
AOR |
95% CI |
p-Value |
|
Gender |
||||||
|
Male |
1.427 |
1.40- 1.44 |
<0.001 |
|
||
|
Female |
Reference |
|
|
|
||
|
Age group |
||||||
|
<5 |
Reference |
Reference |
||||
|
5-14 |
0.699 |
0.62-0.78 |
<0.001 |
0.758 |
0.65-0.87 |
<0.001 |
|
15-24 |
1.487 |
1.33-1.65 |
<0.001 |
1.093 |
0.96-1.23 |
0.152 |
|
25-44 |
2.146 |
1.93-2.37 |
<0.001 |
1.111 |
0.98-1.25 |
0.078 |
|
45-64 |
3.113 |
2.81-3.44 |
<0.001 |
1.627 |
1.45-1.82 |
<0.001 |
|
≧65 |
6.703 |
6.05-7.41 |
<0.001 |
3.803 |
3.39-4.26 |
<0.001 |
|
Intentionality of injury |
||||||
|
Unintentional |
Reference |
Reference |
||||
|
Intentional |
1.983 |
1.90-2.07 |
<0.001 |
2.848 |
2.68- 3.02 |
<0.001 |
|
Intentional unknown |
2.120 |
1.94-2.31 |
<0.001 |
1.981 |
1.77- 2.21 |
<0.001 |
|
Low-income |
||||||
|
Without |
Reference |
Reference |
||||
|
With |
2.303 |
2.19-2.41 |
<0.001 |
2.472 |
2.29- 2.66 |
<0.001 |
|
Catastrophic illness |
||||||
|
Without |
Reference |
Reference |
||||
|
With |
3.432 |
3.36-3.50 |
<0.001 |
3.561 |
3.46- 3.66 |
<0.001 |
|
Psychiatric history |
||||||
|
Without |
Reference |
Reference |
||||
|
With |
1.656 |
1.62-1.68 |
<0.001 |
1.555 |
1.52- 1.59 |
<0.001 |
|
CCI_R |
1.060 |
1.05-1.06 |
<0.001 |
1.076 |
1.07- 1.08 |
<0.001 |
Adjusted OR = Adjusted odds ratio: Adjusted variables listed in the table, CI = confidence interval, AOR:Adjusted Odds Ratio, Nagelkerke R-square = 0.172 (Overall), 0.167 (Male), 0.178 (Female), Some patients didn't provide the information of cause and intentionality of injury
- Discussion:
7.1 I think the discussion about monkeys etc. is unnecessary. It swells the discussion. Only studies, theories, etc. of humans should be presented. After all, it's no secret that more men than women have accidents. This has certainly been discussed elsewhere. It should be revised.
A7.1 Thank you for your comment. We agree with your viewpoint and have deleted the discussion about monkeys, as follows:
Page 10 Line 299 to 320:
Why do men have a greater risk of dying from “injury” than women? At present, it is generally believed that because of socialized relationships, men have engaged in more dangerous behaviors than girls since they were young and are less supervised by people who might protect them (to keep little boys from being harmed). Based on the current theory, attributing all risk-taking behaviors between men and women to “gender differences” is the only explanation that can be explained [29]. Observations on other primates and even other mammals show that “sex differences” are similar in injury and death patterns. The risk-taking behavior of primates is similar to that of humans [30-33]. Mammals are similar to human toy preferences, indicating that differences in toy choices may reflect gender differences in activity preferences, rather than the main caused by the process of socialization [31].
Round 2
Reviewer 1 Report
The authors have made satisfactory changes in line with the earlier review comments.
Author Response
Thanks for your review.
Reviewer 3 Report
Table 1: "age" not "ge"
Author Response
Thanks for your suggestion. We have corrected it.